# A Systematic Review and Meta-Analysis of Fall Prevention Programs for Pediatric Inpatients

**DOI:** 10.3390/ijerph18115853

**Published:** 2021-05-29

**Authors:** Eun-Joo Kim, Geun-Myun Kim, Ji-Young Lim

**Affiliations:** 1Department of Nursing, Gangneung-Wonju National University, Gangneung 26403, Korea; kimeju@gwnu.ac.kr (E.-J.K.); gmkim@gwnu.ac.kr (G.-M.K.); 2Department of Nursing, Inha University, Incheon 22212, Korea

**Keywords:** fall, meta-analysis, pediatric, prevention

## Abstract

Falls account for a high proportion of the safety accidents experienced by hospitalized children. This study aims to analyze the contents and effects of fall prevention programs for pediatric inpatients to develop more adaptable fall prevention programs. A literature search was performed using PubMed (including Medline), Science Direct, CINAHL, Embase, and Cochrane. We included articles published from the inception of each of the databases up to 31 March 2019. A total of 1725 results were reviewed according to the inclusion and exclusion criteria, and nine studies were selected. Data were analyzed using descriptive statistics and the Comprehensive Meta-Analysis program. Four of the nine studies divided their participants into a high-risk fall group and a low-or medium-risk fall group, and all studies used a high-risk sign/sticker as a common protocol guideline for its high-risk fall group. The odds ratio of 0.95 (95% Cl 0.550–1.640) for the fall prevention program in seven studies was not statistically significant. To develop a standardized fall prevention program in the future, randomized control trial studies that can objectively measure the fall rate reduction effect of the integrated fall prevention program need to be expanded.

## 1. Introduction

The safety management of patients has been considered an important factor for enhancing the quality of health care worldwide. Countries around the world are promoting patient safety by implementing healthcare institution accreditation systems, and South Korea has also launched a more systematic patient safety management effort by enacting the Patient Safety Act in 2016. However, according to a 2016 patient safety incident report, the total number of fall incidents was 5562, of which 254 involved children [1]. In particular, falls accounted for 24% of pediatric safety incidents [1]. Children are highly curious and impulsive, while their physical functions and cognitive growth are still under development; hence, they have poor judgment and lack the ability to protect themselves during dangerous situations. As a result, children have a high risk of facing such accidents and have a particularly high risk of falling due to the stage of development they are in and their ambulatory ability [2].

A study that analyzed falls among pediatric inpatients reported a higher rate of falls among children aged one to three years and observed that most falls occurred from the inpatient beds and in the presence of the caregiver [3]. A study on 26 child hospitals in the US reported that 0.4–3.8 cases of pediatric fall incidents occur per 1000 days in the hospital [3]. The rate in Korea is much higher, at 0.63–2.45 cases per 1000 pediatric inpatients, highlighting the gravity of the risk of falling among pediatric inpatients in Korea [4].

Aggressive management to prevent falls among pediatric inpatients is crucial, as falls induce injuries such as abrasion (12.5%), fracture (12.5%), and hematoma (37.5%), and even lead to disability in some cases in children [5]. To this end, the importance of the appropriate evaluation, safe environment, and fall prevention education is emphasized. Previous studies have further suggested that the contents of fall prevention education should include learning about various preventive behaviors and methods and that the education programs should target not only high-risk pediatric patients but also healthcare providers and caregivers of pediatric inpatients to promote attention and interest [6]. In particular, pediatric inpatients were found to be involved in falling incidents even when their caregiver is present, which suggests that inpatient education alone cannot effectively reduce falls, further highlighting the importance of fall prevention education for caregivers such that caregivers can stay around to protect children at all times [2].

Studies that have implemented fall prevention programs have utilized audiovisual materials such as posters and videos, bulletins, and one-to-one education for children’s caregivers using PowerPoint presentations and videos. Park [2] developed and delivered a case-specific fall prevention education program one-to-one and observed significant improvements in fall-related knowledge, attitude, and fall prevention behaviors among caregivers of pediatric inpatients. Park and Ju [4] delivered fall prevention education using a leaflet and reported that fall-related knowledge and fall prevention behaviors significantly increased among the caregivers of pediatric inpatients. However, both studies delivered the educational content for only two to three sessions and for less than 15 min per session, thus lacked the evidence to ascertain whether the developed programs can effectively prevent falls in long-term pediatric inpatients. Further, fall prevention programs comprise an array of components, including views on falls as safety issues, age-specific characteristics, and factors that hinder fall prevention, which undermine their efficiency. As shown here, despite the high perceived need for fall prevention for pediatric inpatients, existing fall prevention education programs are designed only as short-term programs, calling for an evidence-based framework to identify effective fall prevention programs.

Promoting quick recovery and maintaining good quality healthcare by preventing secondary impairments and injuries is important. This can be ensured only by preventing falls among pediatric inpatients in the first place, which requires evidence-based and rational fall prevention interventions that ensure increased effectiveness. A comprehensive review of studies on fall prevention programs for pediatric inpatients needs to be conducted to identify the features and factors related to fall prevention programs and, consequently, develop more robust programs. Thus, this study aimed to conduct a literature review of existing studies on fall prevention programs for pediatric inpatients, based on which we attempted to propose a future direction for the development of effective fall prevention programs. The eventual development of such programs among pediatric inpatients would contribute to improving nursing practice.

## 2. Materials and Methods

This study conducted a secondary data analysis as part of a systematic review and meta-analysis of studies on fall prevention programs for pediatric inpatients.

### 2.1. Procedures

#### Literature Search Databases

The literature search was conducted on PubMed (including Medline), Science Direct, CINAHL, Embase, and Cochrane. The search period included articles published from the inception of each of the databases up to 31 March 2019. The articles published in English were selected because other languages were not assessable.

### 2.2. Search Terms

Search terms were: (child or pediatric or pediatric or children), (inpatient or hospitalized), (fall or fall down or accident fall or fall risk), prevention, (intervention or program), and randomized control trials (RCTs). The search period was set to the earliest year of publication supported by each database to the date of search in 2019.

### 2.3. Literature Selection Process

#### 2.3.1. Literature Inclusion and Exclusion Criteria

From the search results, RCTs and observational studies that investigated the effects of fall prevention programs for pediatric inpatients (age cutoff, <18 years) were selected in the first round. The following cases were excluded from the final analysis: duplicate studies across databases using ENDNOTE, simple surveys that did not examine the effects of the program, studies on falls at home and in the community, and those that were not conducted on pediatric inpatients.

#### 2.3.2. Identification of Final Data for Analysis

Two professor-level researchers finalized the selection of studies according to the inclusion and exclusion criteria described above, and any disagreements were resolved based on the advice of a third professor-level researcher. The entire process was carried out in adherence with the Cochrane guidelines [7] and the details of the process have been delineated in a PRISMA protocol [8] (Figure 1). The specific data selection process is as follows. First, the literature search generated 1725 results and, after deleting 1667 duplicate and irrelevant studies, the titles and abstracts of 1626 studies were reviewed against the inclusion and exclusion criteria. After this step, 12 studies were found to be non-inpatient child and 20 were evaluated to have non-relevant studies, resulting in nine studies being included in the final analysis (Figure 1). Independent reviewers extracted data using a standardized form that included details about authors, year of publication, country where the study was conducted, study design, methods, sample size, intervention, results, and outcome data. To assess the quality and risk of bias of the study, we used the Cochrane risk of bias instrument for Randomized Control Trials (RCT) and the risk of bias assessment tool for Non- RCT.

### 2.4. Ethical Consideration

This study was reviewed by the Human Research Ethics Committee of Gangneung-Wonju National University [GWNUIRB-2019-01-21]. The committee decided that the present study was exempt from ethical approval.

### 2.5. Data Analysis

The results were statistically analyzed using the Comprehensive Meta-Analysis software version 3.0 (Boistat, Englewood. NJ, USA). To conduct a meta-analysis, the effect sizes were calculated based on the sample size, odds ratio, fall rates, percentage of fall events, and statistical significance of each experimental and comparator group per each study. We analyzed dichotomous data as odds ratios with 95% confidence intervals (CIs) and continuous data as mean difference with 95% CIs. Heterogeneity was evaluated using the I^2^ test. The publication bias was assessed by examining a funnel plot.

## 3. Results

### 3.1. Features of the Studies Selected for Analysis

The characteristics of the included studies are described in Table 1. The studies selected for analysis were published between 2007 and 2017. Except for a study published in Korea and another in Singapore, all studies were published in the United States. Five studies were quality improvement (QI) or project studies, two were experimental studies, and two were retrospective chart review studies. Regarding the experimental group and control group, two studies were conducted on caregivers, while the remaining studies were conducted on pediatric inpatients in the pediatric ward or the Pediatric Intensive Care Unit to measure the number of fall events, fall rate, and fall risk.

### 3.2. Specific Contents of the Pediatric Inpatient Fall Prevention Programs in the Selected Studies

Table 2 provides details of the fall prevention programs investigated in the nine selected studies. Four out of nine studies divided the patients into a high-risk fall group and a low-or medium-risk fall group to apply the corresponding protocol. Using the high-risk sign/sticker was a common protocol guideline in all studies concerning the high-risk fall group. Cooper and Nolt, Hill-Rodriguez et al., Kim et al., Neiman et al., Rouse et al., and Stubbs and Sikes [9,10,11,14,16,17] recommended the signs to be shown on the bed and outside the ward, while Lee et al. [13] recommended that patients wear a green tag on the wrist or ankle. The next guideline was close observation. Neiman et al., and Murray et al. [11,15] stressed the importance of hourly rounding and one-to-one observation, while Cooper and Nolt, Hill-Rodriguez et al., and Rouse et al. [9,10,14] suggested placing patients close to the registered nurse (RN) station and leaving the room door open. The next important guideline was to assist with patients’ ambulation. The guidelines suggested frequently checking and safely supporting the patients. The general protocol applied to the low-and moderate-risk group included educating patients and their families, using low beds, always leaving the bed rails up with locked breaks, and never leaving the child alone.

### 3.3. Meta-Analysis Results

Figure 2a presents a forest plot of the seven studies. The studies showed very low homogeneity (I^2^ = 70.3) and showed an odds ratio of 0.95 (95% Cl 0.550~1.640); hence, it was not significant at a z-value of −0.184 (*p* = 0.854). Figure 2b presents the forest plot of only four studies in which the number or the rate of inpatient falls is the outcome. The I^2^ of these four studies was 32.13, showing a random effect, while the odds ratio of the fall prevention programs was 0.561 (95% Cl 0.333~0.945, z-value = −2.173, *p* = 0.030).

In other words, the fall-prevention programs for inpatients have been effective in reducing the fall rate. It was confirmed that there was no publication bias because the corrected effect size and the original effect size were the same by adding a study to be symmetrical through funnel plot, Duval and Tweedie’s trim and fill method (Figure 3).

## 4. Discussion

A systematic review and meta-analysis of studies on fall prevention programs for pediatric inpatients were conducted. Of the 1725 search results, nine studies were selected for analysis, of which seven were meta-analyzed. Of the nine selected studies, one was conducted in Korea, while eight were conducted in foreign countries. Further, five studies were QI or project studies, two were experimental studies, and the remaining two were retrospective chart review studies, showing that experimental studies to measure the effects of fall-prevention programs are relatively scarce. In four studies that accurately presented the fall rate value, the fall prevention program was found to be effective.

Experimental studies on fall prevention programs are lacking, despite the gravity of fall risks among pediatric inpatients, as shown by the high incidence of 0.63–2.45 cases of falls per 1000 pediatric inpatients [4]. This suggests that there is little evidence supporting the effectiveness of fall prevention programs currently administered in clinical practice. In other words, most studies on falls in pediatric inpatients were QI or project studies that applied the protocol used in the corresponding clinical institutions. This is due to the difficulty of implementing RCT in the clinical field. In particular, seven of the nine studies were conducted in the United States.

This indicates that the current approaches to prevent falls in pediatric inpatients are organization-specific and individual approaches that cannot be conclusively determined to have been based on objective evidence and have little evidence to validate their effectiveness. Thus, developing a standardized protocol for preventing falls among pediatric patients is needed. This is a limitation of this study. Therefore, it seems that research such as retrospective chart reviews should be accumulated in the future. However, the lack of such evidence suggests that a standardized program is needed to prevent falls in hospitalized child patients. In addition, most of these studies were conducted by assessing the risk of falling and applying institutional protocols accordingly. Although intervention by fall risk level seems to be desirable, it needs to be approached while considering the characteristics of the children’s developmental stage.

Regarding the specific contents of pediatric inpatient fall prevention programs in the selected studies, only four of the nine studies classified fall risk into high risk and low risk [11,12,15,16]. All nine studies emphasized the importance of the high-risk group. The common protocol for the high-risk group included several components: (a) Using stickers or signs to indicate the patients’ fall risk; (b) performing closed observation, such as by placing the patients in a room close to the RN station or leaving the door open; and (c) assisting with ambulation by checking in on the patients often.

Some studies also advanced a general protocol for the low-risk group. This protocol included educating the patients and their families, leaving the bed guards up with locks, and having caregivers stay around the patient at all times. While the children identified to be at high risk of falling are the prioritized targets for interventions, children with moderate or lower risk should also be given appropriate intervention. It is also necessary to examine the fact that most studies primarily applied prevention behaviors to the high-risk group as opposed to classifying children based on their risk level. In the case of low-risk groups, family members were particularly included in the intervention. This is because including not only nurses but also patients and their families was shown to be effective when considering fall prevention programs according to each risk level. In other words, to reduce the incidence of falls, a team approach between the child, family, and professional is also necessary [18].

Based on the results of the meta-analysis, the four studies showing significant results emphasized regular and frequent rounding (hourly rounding and 1:1 observation), and the composition of a safe hospitalization environment (bed brake, bed in low position, side rail up). To support children’s ambulation and maintain a safe environment, families were included in the fall assessment, and family and staff education for fall prevention were also included. For more active management, the fall risk level of children was determined by using the fall risk assessment tool. This is consistent with the results of Benning and Webb [18] who use evidence-based tools to successfully apply a children’s fall prevention management program. They emphasize that both families and nurses should actively participate in fall prevention activities. For fall prevention activities to be successful, it is necessary to accurately evaluate the child’s fall risk using evidence-based tools and to determine the level of fall prevention management activities. Accurate reporting is needed to evaluate the incidence of pediatric falls. Future implications for practice involve collaboration with patients, families, and interprofessional team members to co-design fall prevention improvement activities in all hospital settings.

Most of the studies that classified fall risk into high risk and low risk used the Humpty Dumpty Fall Scale (HDFS) to assess children’s fall risk. The HDFS is widely used in clinical settings but was reported to have markedly low specificity, as evidenced by the fact that 80% of children are identified to be in the high-risk group using the scale [19]. A meta-analysis of the instruments that assess fall risk in children, such as HDFS, Graf-PIF, and I’m Safe, found that they have low sensitivities, as they do not feature detailed score classifications, with many items also measured based on the nurses’ personal judgments [11]; hence, the sensitivity of the instruments that assess fall risk is an important factor. However, it is currently lacking in the existing scales.

This shows that the priority in fall-prevention needs to be the development of an instrument that can objectively and accurately assess fall risk in children by considering the unique characteristics that make them risk-prone. Only once such an instrument is developed can children’s fall risks be categorized into the high-risk or the relatively low-risk group and risk-specific fall prevention programs be suitably administered. Once children’s risk level is determined, the high-risk group can receive a careful intervention by nurses, while the low-risk group can receive an intervention using various approaches, including caregiver education.

In addition, in the case of older adults, multifactorial interventions using assistive devices such as bed alarm devices, walking aids, and hip protectors to prevent falls and improve the environment around the subject to make it safe are more effective than single interventions [20,21]. In the case of children, in consideration of their developmental stages, multi-factorial intervention methods using auxiliary tools such as bed alarm devices and wireless devices need to be considered.

In the meta-analysis of the seven studies whose statistics were available, no significant effects were observed. In the four studies that suggested the exact fall rate, the fall prevention program had an effect. This study included articles from only a few countries; hence, the generalizability of the results is limited. However, the studies on fall programs for inpatient children are scarce and most of them are project-specific. Moreover, very little RCT research has been conducted on this topic. Therefore, expanding the application of effective fall prevention intervention programs is essential, in addition to expanding fall prevention research using more sophisticated research designs. Ultimately, a standardized fall prevention program based on results of this study should be developed.

## 5. Conclusions

This study attempted to examine the existing findings of previous studies on fall prevention programs through a systematic review and to propose a direction for developing effective fall prevention programs based on this examination. Most of the studies were project studies, showing that the current literature does not present enough objective evidence. Only four studies divided the patients by risk level, and the instruments used to assess fall risk were also limited. This suggests that a fall risk assessment instrument needs to be developed. Furthermore, to prevent falls among pediatric inpatients, it is necessary to develop an evidence-based, standardized program and assess its effectiveness.

Most of the studies included in the analysis in this study were conducted in the United States and many constituted QI projects rather than an RCT. Additionally, since the number of analyzed studies is very small, it is difficult to generalize and more studies need to be accumulated.

## Figures and Tables

**Figure 1 ijerph-18-05853-f001:**
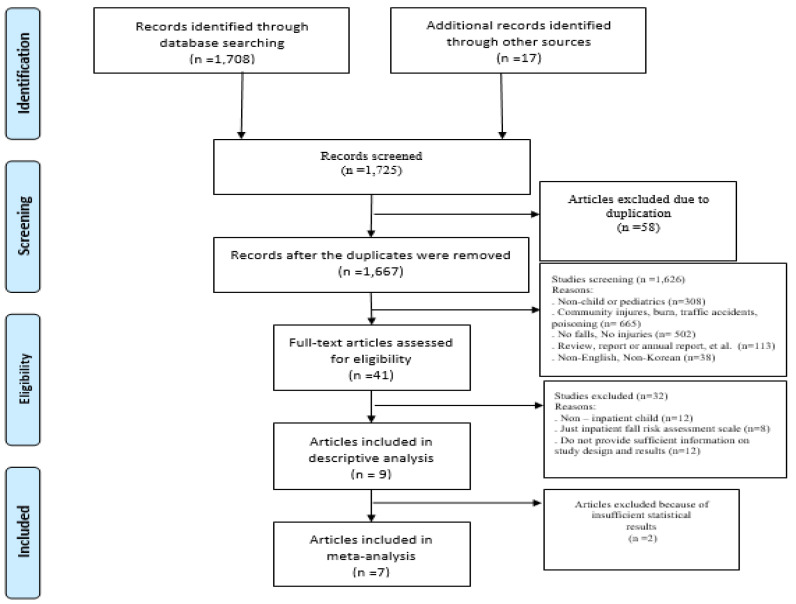
Flow diagram depicting study selection.

**Figure 2 ijerph-18-05853-f002:**
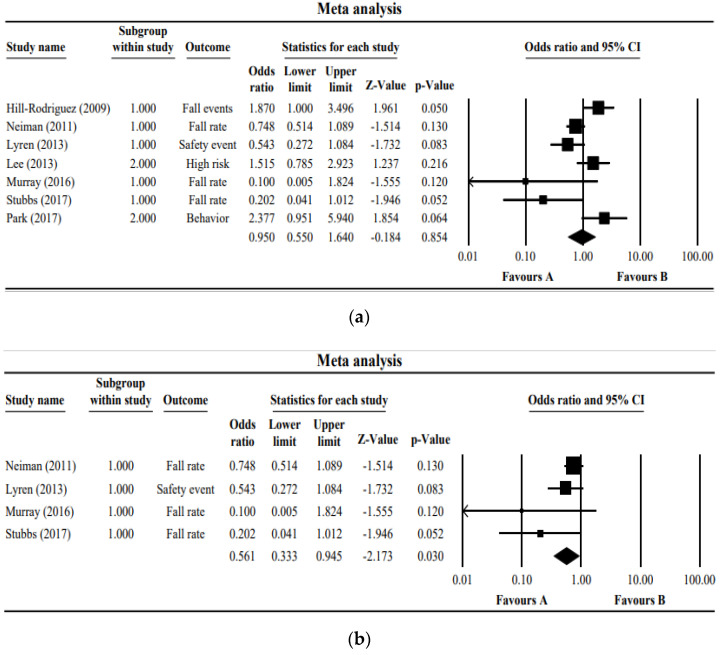
(**a**) Forest plot of the studies included in the meta-analysis. (**b**) Forest plot of the studies measuring the number or the rate of inpatient falls as an outcome. The boxes in the graphs show the effect estimates from the single studies, while the diamond symbol shows the pooled result.

**Figure 3 ijerph-18-05853-f003:**
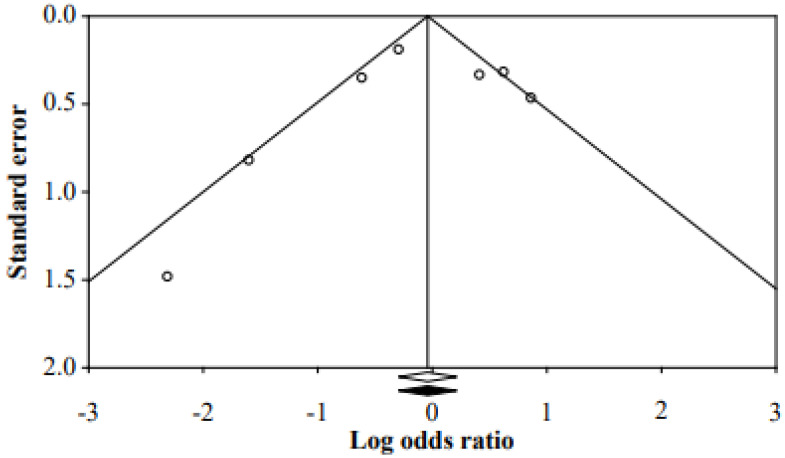
Funnel plot of the studies.

**Table 1 ijerph-18-05853-t001:** General characteristics of the selected studies.

Author (Year)	Nation	Design	Setting	Experimental (n or Period)	Control (n or Period)	Program	Intervention Period	Result
Cooper and Nolt (2007) [9]	USA	Project-a prospective descriptive chart review	General pediatric unit 0–21 years	Not described	Not described	Pediatric Fall Prevention Program for inpatients or outpatients	January–June 2006	Not described in detail
Hill-Rodriguez et al. (2009) [10]	USA	Matched case-control design	In-patient units, PICU ^1^, CICU ^2^	153	153	Humpty Dumpty Procedure: Low/High program	2005–2006	Fall events: Odds ratio = 1.87; confidence interval = 1.01, 3.53 (*p* = 0.03)
Neiman et al. (2011) [11]	USA	Retrospective case-control study	Inpatient encounter	59	177	I’M SAFE fall risk tool/prevention/evaluation	January 2004–September 2005	Decreased fall rate = 0.67/1000 patient days >0.51/1000 (*p* = 0.015)
Lyren et al. (2013) [12]	USA	QI project	Children’s hospitals	45	127	Ohio Children’s Hospital Association	January 2010–October 2012	Decreased severe safety event (70/127->18/45)
Lee et al. (2013) [13]	Singapore	Experimental study	Pediatric wards	30 (caregivers)	None	The JBI Practical Application of Clinical Evidence System (PACES) and Getting Research into Practice (GRiP) Programmes	March–June 2011	(1) The fall risk preventative interventions and high-risk fall event did not differ between the experimental and control groups (*p* = 0.110) (2) The behavior of patients at risk of fall did not differ between the experimental and control group (*p* = 0.039)
Rouse et al. (2014) [14]	USA	QI project	Pediatric unit	Not described	Not described	Patient Falls Safety protocol	June 2011	Not described
Murray et al. (2016) [15]	USA	QI project	Pediatric ward/PICU ^1^	January–June 2016	January–June 2015	Plan-Do-Study-Act (PDSA)	January–June 2016	Decreased fall rate 4.5% per 1000
Stubbs and Sikes (2017) [16]	USA	QI project	Inpatient pediatric neuro rehabilitation center	2014	2009	PDSA Method: Interdisciplinary intervention-green light, green light	2010–2014	(1) Decreased fall rate 8.84/1000 patient days, 1.79/1000 patient days (χ^2^ = 17.23, *p* = 0.0001) (2) Decreased falls with caregivers (χ^2^ = 6.25, *p* = 0.012)
Park and Ju (2017) [4]	Korea	Non-equivalent control group, non-synchronized design	Pediatric ward	31 (caregivers)	31 (caregivers)	Pediatric Fall Prevention Education: A leaflet and picture book	August–October 2013	(1) There was a difference in fall-related knowledge between the experimental group and control group (t = −3.05, *p* = 0.048) (2) There was no difference in the preventive behaviors for patients at risk of falls between the experimental group and control group (t = −1.91, *p* = 0.065)

^1^ Pediatric intensive care unit. ^2^ Cardiac incentive care unit.

**Table 2 ijerph-18-05853-t002:** The detailed contents of the inpatient fall prevention programs.

Author (Year)	Program	High Risk Protocol	Low Risk Protocol
Cooper and Nolt (2007) [9]	Pediatric Fall Prevention Program for inpatients	- Humpty Dumpty sign on patient’s door - Humpty Dumpty sticker on patient’s chart - Meditech bulletin boards updated “Fall Risk” and dated - Room closer to RN station - Consider utilizing sitters, volunteers, family - Assist with toileting At frequent, scheduled intervals - Provide assistive devices to steady gait - Request order for physical therapy as appropriate - Request order for restraints as appropriate - Document: documented in their care plan for high-risk group	- Use cribs for all patients <3 years old - Encourage skid-resistant shoes/slippers - Assist unsteady patient with ambulation - Manage to improve mobility - Keep bed in the lowest position, brakes on - Eliminate clutter in the room - Keep call light within reach and answer promptly - Place articles (glasses, hearing aids, mobility aids)
Hill-Rodriguez et al. (2009) [10]	Patient Falls Safety Protocol	- Identify patient with a “Humpty Dumpty sticker” on the patient, in the - bed, and in the patient chart - Educate patient/parent of fall protocol precautions - Check patient with ambulation - Accompany patient with ambulation - Developmentally place patient in appropriate bed - Consider moving patient closer to nurse’s station - Assess need for one-to-one supervision - Evaluate medication administration times - Remove all unusual equipment out of the room - Protective barriers to close off spaces, gaps in the bed - Keep door always open unless patient is directly attended - Document in nursing narrative teaching and plan of care	- Orientation to room - Bed in low position, brakes on - Siderail *2 or 4 up, assess large gaps - Nonskid footwear - Assess elimination needs, assist as needed - Call light within reach, educate patient/family - Environment clear of unusual equipment, furniture, and hazards - Assess for adequate lighting, leave night light on - Educate patient and parent - Document fall prevention
Neiman et al. (2011) [11]	I’m SAFE fall Prevention Program	- I’m Safe Fall Assessment tool by EMR, hourly rounding, one to one - observation - Safe room set up (bedside signage, bed brake, bed in low position, side rail up)	Low-risk intervention - Family education - Bed in low position, side rail up, bed brakes on, clutter in room minimized - Moderate risk intervention - Assist with activity/mobility - Periodic assessment of elimination - Periodic orientation
Lyren et al. (2011) [12]	Collaborative Organizational Framework-High Reliability Implementation	The error prevention task force The leadership methods task force The cause analysis task force The lessons learned task-communication, risk management All organizations have developed mechanisms to routinely share safety stories The safety governance task forces	
Lee et al. (2013) [13]	The JBI Practical Application of Clinical Evidence System (PACES)	- Perform reinforcement and PFE ^1^ on fall prevention Please do not leave your child alone Please raise and security lock both bed rails The green wrist tag on your child’s wrist and ankle - Develop a poster on fall prevention	
Rouse et al. (2014) [14]	Patient Fall Safety Protocol	Similar to Cooper and Nolt’s (2007) protocol	
Murray et al. (2016) [15]	Fall Risk Assessment, prevention program	Plan-Do-Study-Act (PDSA)-6-bed ward, PICU, 0–18 months, Fall Risk Assessment, Prevention program/HDFS (administered once a shift/family) and patient education, sign, orientation to the unit, environment safety, patient rounding hourly (high risk)	
Stubbs and Sikes (2017) [16]	PDSA method: interdisciplinary intervention—red light, green light	Red Green light Interdisciplinary care involving physical therapist, nurse manager, educator Family training session, red-green light Staff education/nursing staff education	
Park and Ju (2017) [4]	Pediatric fall prevention education	Pediatric fall prevention education: Leaflet and picture book	

^1^ PFE: Patient and Family Engagement.

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
