# Peer review of "A Systematic Review and Meta-Analysis of Fall Prevention Programs for Pediatric Inpatients"

_ijerph, 2021, doi:10.3390/ijerph18115853_

Round 1
Reviewer 1 Report
Comments to "A systematic review and meta-analysis of fall prevention programs for pediatric inpatients":
- I don't think it is necessary to include ": (1) Background :" ant others in the abstract. IJERPH shows us a guide with these statements.
- The abstract must initially include a phrase that focuses the reader on the subject of the study. That is, a brief description of the current or historical topic.
- What is the reason for not including other databases such as Scopus or WoS? These also include scientific articles related to prevention programs for pediatric inpatients.
- You must correct the citations in the text. They are numbered in order.
- The introduction is adequate but not quite cited.
- You must include a section 2 in which you describe the most prominent concepts in the study. That is, why do you use: (child or pediatric or pediatric or children), (inpatient or hospitalized), (fall or fall down or accident fall or fall risk), prevention, (intervention or program), and randomized control trials (RCTs )? Why not "adolescent" "boy" "girl", for example?
- It is convenient to note how the studies are approached with the downton scale.
- How can the data mining improve this work? It would be appropriate to comment on it.
- What are the limitations that you have found in this study?
Author Response
Thank you for your valuable academic comments for our manuscript. We are trying to do our best to improve quality of manuscript according to your comments. More details, we marked in manuscript in red.
- I don't think it is necessary to include ": (1) Background :" ant others in the abstract. IJERPH shows us a guide with these statements.
-> Yes, according to your comments, we modified abstract and marked in red.
- The abstract must initially include a phrase that focuses the reader on the subject of the study. That is, a brief description of the current or historical topic.
-> Thank you for your valuable comments. According to your comments, added sentences in abstract.
- What is the reason for not including other databases such as Scopus or WoS? These also include scientific articles related to prevention programs for pediatric inpatients.c
-> I think Medline and Embase contain most of the index journals from Scopus or WoS.
- You must correct the citations in the text. They are numbered in
-> Yes, We did correction.
- The introduction is adequate but not quite cited.
-> We added several references in introduction part.
- You must include a section 2 in which you describe the most prominent concepts in the study. That is, why do you use: (child or pediatric or pediatric or children), (inpatient or hospitalized), (fall or fall down or accident fall or fall risk), prevention, (intervention or program), and randomized control trials (RCTs )? Why not "adolescent" "boy" "girl", for example?
-> Since this study is for inpatient pediatric, it covers ages 0-18 or 21 years old.
In addition, I think that it can be excluded because it is not common to express inpatients in terms of boy or girl.
- It is convenient to note how the studies are approached with the downton scale.
-> Thank you for your valuable comments. We used PRISMA tool for approaching studies. Please see figure 1.
- How can the data mining improve this work? It would be appropriate to comment on it.
-> Thank you for your valuable comments. In this study, we try to find what research trend regarding pediatric falls was from reviewing previous studies. Therefore, we consider data mining method for next step study.
- What are the limitations that you have found in this study?
-> According to your comments, we revised as follows:
Most of the studies included in the analysis in this study were conducted in the United States, and there were many cases that took the form of a QI project rather than an randomized control trial. Also, since the number of analyzed studies is very small, it is difficult to generalize and more studies need to be accumulated.

Reviewer 2 Report
This systematic review included 9 studies and summarized the effects of fall prevention programs in pediatric inpatients. Results are not conclusive due to the different design of studies and different programs. Overall, the manuscript is well written. There are strengths and weaknesses in this study. My following comments include a summary of weaknesses and offer suggestions for the authors' consideration.
- In Abstract, numbers in front of each section (e.g., (1) background, (2) methods) can be deleted unless it is required by the journal. The conclusion section may provide recommendation on what program(s) would be more effective.
- The introduction has provided relevant information to support the study objective. The authors mentioned the fall-related injuries in the third paragraph. However, the rates of injuries were not provided. If possible, please provide data on the rates of common injuries as these data will support the importance of fall prevention in pediatric patients.
- In the methods, search terms were briefly described. Please also provide the key words. In addition, Figure one is not the standardized flow chart. Please use the template provided by PRISMA.
- In data analysis, please provide more details on the “Comprehensive Meta-Analysis program”, including what software was used and how the data were analyzed.
- The results reported that one study was from Korea. However, Table 1 shows that studies were from the USA and Singapore. Please clarify.
- In Table 1, please add the setting (e.g. pediatric ward, ICU), which can be under the author or in a separate column. In addition, the summary of results is not consistent in Table 1 (e.g. fall events, fall rate, odds ratio). Please provide the key findings of each study such as decreased fall rate/events (similar as the last study) or no significant changes.
- In this study, gender differences in fall rates or response to the interventions were not described in results, while it was briefly mentioned in discussion. Please provide more results on gender differences from the reviewed articles if available.
- In the discussion, interpretation of results needs to be cautious due to the small number of articles (with 5 qualitative studies). This section discussed the fall risk classifications and importance of developing the fall risk assessment instruments. However, discussion on the effectiveness of fall prevention programs is lacking. In addition, are there any barriers or facilitators to the fall prevention programs that may affect the effectiveness? This information is important for the future design of interventions.
- Are there any limitations of this study? Please include them in the discussion if any.
- In the conclusions, the authors state that it is necessary to develop an evidence-based, standardized program. However, details of the recommendation were not provided. These can be drawn from the key findings of the intervention programs (see my comment #8).
Author Response
Thanks you for your valuable academic comments. We are trying to do our best to improve the quality of manuscript. More details, we marked in manuscript in red.
- In Abstract, numbers in front of each section (e.g., (1) background, (2) methods) can be deleted unless it is required by the journal. The conclusion section may provide recommendation on what program(s) would be more effective.
-> Yes, according to your comments, we modified abstract and marked in red.
- The introduction has provided relevant information to support the study objective. The authors mentioned the fall-related injuries in the third paragraph. However, the rates of injuries were not provided. If possible, please provide data on the rates of common injuries as these data will support the importance of fall prevention in pediatric patients.
-> Thank you for your valuable comments. According to your comments, added rate information in third paragraph.
- In the methods, search terms were briefly described. Please also provide the key words. In addition, Figure one is not the standardized flow chart. Please use the template provided by PRISMA.
-> Thank you for your valuable comments. According to your comments, revised figure1.
- In data analysis, please provide more details on the “Comprehensive Meta-Analysis program”, including what software was used and how the data were analyzed.
-> Thank you for your valuable comments. According to your comments, added.
- The results reported that one study was from Korea. However, Table 1 shows that studies were from the USA and Singapore. Please clarify.
-> Thank you for your valuable comments. According to your comments, revised table 1 and rechecked.
- In Table 1, please add the setting (e.g. pediatric ward, ICU), which can be under the author or in a separate column. In addition, the summary of results is not consistent in Table 1 (e.g. fall events, fall rate, odds ratio). Please provide the key findings of each study such as decreased fall rate/events (similar as the last study) or no significant changes.
-> Thank you for your valuable comments. According to your comments, revised table 1 and rechecked.
- In this study, gender differences in fall rates or response to the interventions were not described in results, while it was briefly mentioned in discussion. Please provide more results on gender differences from the reviewed articles if available.
-> In the analysis in this study, we did not focus on gender differences. The discussion was focused on the composition of the entire program and the results derived.
- In the discussion, interpretation of results needs to be cautious due to the small number of articles (with 5 qualitative studies). This section discussed the fall risk classifications and importance of developing the fall risk assessment instruments. However, discussion on the effectiveness of fall prevention programs is lacking. In addition, are there any barriers or facilitators to the fall prevention programs that may affect the effectiveness? This information is important for the future design of interventions.
-> Thank you for your valuable comments. According to your comments, we revised.
- Are there any limitations of this study? Please include them in the discussion if any.
-> Thank you for your valuable comments. According to your comments, we revised.
- In the conclusions, the authors state that it is necessary to develop an evidence-based, standardized program. However, details of the recommendation were not provided. These can be drawn from the key findings of the intervention programs (see my comment #8).
-> Thank you for your valuable comments. According to your comments, we revised.

Reviewer 3 Report
This paper uses a systematic review and meta-analysis to search fall prevention programs for pediatric inpatients in order to propose a direction for developing effective fall prevention programs. However, some research contents still need to be modified.
- In the Identification of Final Data for Analysis in page3, the meta-analysis method this research applies is used to find internal connections between fall rate of pediatric inpatients and some influence factors. But this paper make analyses according to nine studies, which seem to be less reliable to prove the conclusion of this paper. In other words, why this nine articles can be used to represent all research.
- Inthe Flow Diagram Depicting Study Selection in page 4, most of articles are excluded according to some criteria. These standards are not described in detail, so the reasons why thousands of articles are gotten rid of instead of being taken into consideration are not enough.
- Figure 2 and Figure 3 in page 9 can be merged into one figure to simplify the results.
- The references this paper quoted need to be sorted in chronological and research type order.
- The biggest problem of this paper is that it looks more like a literature review than a scientific research. It aims to analyze fall prevention programs for pediatric inpatients to develop more adaptable programs. But this paper pays most attention to the existing findings of previous studies than proposing better programs. It is suggested that more further work can be put into effect in this paper.
To sum up, I suggest rejecting this article.
Author Response
The manuscript has been rechecked and the necessary changes have been made in accordance with the reviewers’ suggestions. Our revisions in the manuscript have been marked in red colored text. The responses to all comments have been prepared and attached herewith.
Thank you for your consideration.

Round 2
Reviewer 1 Report
Comments to "A systematic review and meta-analysis of fall prevention programs for pediatric inpatients":
In the first place, I want to inform you that I do not understand the current trend of making a manuscript in a JCR-Q1 in so few pages (considering that MDPI has also reduced the template format, it would be much fewer pages). In this case, there are 12 pages including references, tables and figures, or leaving wide blank spaces like those on page 4. Sometimes, a study can be perfectly synthesized in a few pages, but it is true that most of the cases not so, and even less a systematic review article.
Why do you present data up to 2019? Why not extend to 2020 if the manuscript is prepared in 2021?
In this study, a deeper analysis of the study variables and a greater discussion related to the previous literature analyzed is mainly missing. The conclusions are succinct.
Author Response
Thank you for your valuable comments.
We are trying to do our best for updating the manuscript following your comments.
Please check the attached files to find more details.

Reviewer 2 Report
Thanks for revising the manuscript. The clarity and contents have been improved. I have a few more comments.
- Figure 1 is not very clear. Please increase the figure and text size. Meanwhile, those small arrows can be deleted.
- There are grammar errors and typos in the newly added text. For example, Line 129-132, Table 1 last column (first study), and Line 243-245 (e.g., two "such as" in one sentence), etc. Please carefully check the revision before re-submission.
- The limitations are not clearly discussed. I suggest to add one paragraph of limitations in the Discussion section.
Author Response
Thanks for your valuable comments.
We are trying to do our best for updating the manuscript following your comments.
Please check the manuscript for more details.

Reviewer 3 Report
Comments:
This paper uses a systematic review and meta-analysis to search fall prevention programs for pediatric inpatients in order to propose a direction for developing effective fall prevention programs. The quality of this article has been greatly improved. But there are still some points needed to be modified.
- In the Identification of Final Data for Analysis in page3, the reason why this nine articles can be used to represent all research still not to be interpreted. The author can add explanation here to increase the credibility of the article.
- TheFigure 1. Flow Diagram Depicting Study Selection in page 3 adds some reasons why some studies are excluded. But the picture is not clear enough to see these reasons described.
- The columnwidth of the Table 1 in page 4 seems so close that many sentences have to be divided into rows, which influence reader's perception.
- The Figure 2.and 3. in page 8 present two groups of similar studies ,why not mergethese two figuresinto one to simplify the results.
- The Figure 4. in page 9 draws the Funnel Plot of the Studies according to 7 dots. So the trend of this figure lacks reliability ,to a certain extent.
- In the Discussion in page 9 shows some significant ways to decrease the child`s fall risk, which provides the paper with social value.
To sum up, I suggest accept this article.

Author Response
First of all, we appreciate your valuable comments.
According to your comments, we revised manuscript carefully.
Please check it in attached file.

Round 3
Reviewer 1 Report
Comments to the work "A systematic review and meta-analysis of fall prevention programs for pediatric inpatients":
- It still does not delve into the subject matter nor has he carried out an extensive review of the previous literature.
- The discussions are brief and without an endorsement of the previous literature.
- What literature indicates that there is a gap to review Fall Prevention
Programs for Pediatric Inpatients? Who may be interested?
Author Response
Comments to the work "A systematic review and meta-analysis of fall prevention programs for pediatric inpatients":
- It still does not delve into the subject matter nor has he carried out an extensive review of the previous literature.
In order to carry out this study, the researcher extracted 1,725 documents and read 1,667 abstracts of studies one by one, and extracted articles suitable for the purpose of this study.
- The discussions are brief and without an endorsement of the previous literature.
Based on the opinions of the reviewers, as a researcher, we interpreted the meaning of the results of this study and suggested directions for future research.
- What literature indicates that there is a gap to review Fall Prevention
Programs for Pediatric Inpatients? Who may be interested?
Pediatric falls are a very important topic in child care. In addition, in hospital inpatient nursing, fall prevention is an important index to evaluate the quality of nursing care. Therefore, many people will be interested in this topic, such as graduate students majoring in child nursing, professors who study children's falls, and nursing managers in charge of fall management.
